# Feasibility of using a novel automatic cardiac segmentation algorithm in the clinical routine of lung cancer patients

Robert Neil Finnegan[1], Lucia Orlandini[2,3,4], Xiongfei Liao[2,3,4], Jun Yin[2,3,4]*, Jinyi Lang[2,3,4]*, Jason Dowling[1,5], Davide Fontanarosa[6]

1 Institute of Medical Physics, School of Physics, University of Sydney, Camperdown, New South Wales, Australia, 2 Sichuan Cancer Hospital & Institute, Chengdu, China, 3 Radiation Oncology Key Laboratory of Sichuan Province, Chengdu, China, 4 School of Medicine, University of Electronic Science and Technology of China (UESTC), Chengdu, China, 5 Australian eHealth Research Centre, CSIRO, Herston, Queensland, Australia, 6 Institute of Health Biomedical Innovation, Queensland University of Technology, Kelvin Grove, Queensland, Australia

* scchyinjun@163.com (JY); langjy610@163.com (JL)

**Data Availability Statement:** Data for this work is available on a public repository (Open Science Framework). DOI: 10.17605/OSF.IO/URSGB.

## Abstract

Incidental radiation exposure to the heart during lung cancer radiotherapy is associated with radiation-induced heart disease and increased rates of mortality. By considering the respiratory-induced motion of the heart it is possible to create a radiotherapy plan that results in a lower overall cardiac dose. This approach is challenging using current clinical practices: manual contouring of the heart is time consuming, and subject to inter- and intra-observer variability. In this work, we investigate the feasibility of our previously developed, atlas-based, automatic heart segmentation tool to delineate the heart in four-dimensional x-ray computed tomography (4D-CT) images. We used a dataset comprising 19 patients receiving radiotherapy for lung cancer, with 4D-CT imaging acquired at 10 respiratory phases and with a maximum intensity projection image generated from these. For each patient, one of four experienced radiation oncologists contoured the heart on each respiratory phase image and the maximum intensity image. Automatic segmentation of the heart on these same patient image sets was achieved using a leave-one-out approach, where for each patient the remaining 18 were used as an atlas set. The consistency of the automatic segmentation relative to manual contouring was evaluated using the Dice similarity coefficient (DSC) and mean absolute surface-to-surface distance (MASD). The DSC and MASD are comparable to inter-observer variability in clinically acceptable whole heart delineations (average DSC > 0.93 and average MASD < 2.0 mm in all the respiratory phases). The comparison between automatic and manual delineations on the maximum intensity images produced an overall mean DSC of 0.929 and a mean MASD of 2.07 mm. The automatic, atlas-based segmentation tool produces clinically consistent and robust heart delineations and is easy to implement in the routine care of lung cancer patients.

**Funding:** The author(s) received no specific funding for this work.

**Competing interests:** The authors have declared that no competing interests exist.

## Introduction

Lung cancer is the most common cancer and leading cause of cancer deaths worldwide [1]. Concurrent chemoradiotherapy is a recommended treatment for locally advanced non-small cell lung cancer (NSCLC) and small cell lung cancer (SCLC) [2, 3]. Thoracic radiation exposes the heart to radiation, which can result in damage to the heart. Myocardial infarction is one of proven cardiotoxic effects caused by incidental irradiation of the heart in patients with breast and lung cancer that are treated with radiotherapy [4–6]. There is a dose effect relationship: the higher the dose of the incidental radiation to the heart, the higher the likelihood of a cardiovascular complication, such as a heart attack [7]. These cardiotoxic effects can already occur within a few years after the irradiation [6]. The likelihood of cardiotoxic effects is even higher if there are synergistic cardiac risk factors present such as smoking and hypertension. With the introduction of targeted therapies such as Nivolumab the survival of lung cancer patients will increase [8, 9]. It can be expected that with an increased survival, lung cancer patients with incidental radiation dose to the heart are more at risk to develop radiation induced cardiac toxicity, highlighting the importance of reducing cardiac exposure [10].

A potential factor leading to increased cardiac doses is heart movement during treatment. The displacement of the heart, measured using cone beam computed tomography (CBCT) scans taken during radiotherapy, can be as large as 13 mm [11]. In lung radiotherapy treatment planning, respiratory-gated 4-dimensional CT (4D-CT) is often used to track the movement of the lung tumour [12], and management of respiratory motion is an important step in the workflow of lung radiotherapy. 4D-CT incorporates the patient's respiratory information into a stack of 3D images, such that sequential images at different respiratory phases can be reconstructed [13]. With the 4D images, the tumor excursion range with respiration can be obtained, and a patient-specific internal margin can be included for contouring of the internal tumor volume (ITV) [14].

From these 4D-CT images the movement of the heart can also be captured, and a more realistic planned organ at risk volume (PRV) of the heart can be generated. This PRV of the heart will make it possible to create a radiotherapy plan that will lower the radiation dose to the heart, which will reduce the likelihood of cardiotoxicity and therefore potentially increase patient quality of life and survival. Maximum intensity projection (MIP) and average intensity projection (AIP) images created from the 4D-CT datasets are usually used for treatment planning on a moving target [15], and are currently the most advanced clinical approach to challenge of heart delineation. Manual contouring is a time-consuming process, and is subject to inter-observer variability (IOV) in contouring which may impact the accuracy of dose assessment [16]. In clinical lung cancer radiotherapy, manual contouring of the heart on each phase image may not provide an appropriate solution due to time constraints and observed IOV [17]. It has therefore become paramount to introduce efficient methods to properly manage heart motion during radiotherapy in the routine care of lung cancer patients.

A potential solution lies in the application of automatic segmentation techniques, which have experienced increasing uptake in both research and clinical work. Atlas-based approaches have been used for cardiac delineation in radiotherapy, where they have been shown to both save time in contouring and reduce IOV [18, 19]. We have previously developed an open-source atlas-based framework, designed specifically for use in radiotherapy [20, 21]. This method is flexible, efficient, and accurate. Furthermore, the use of open-source licensing permits distribution between clinics, allowing implementation with local atlas sets and removing the dependence on external datasets. Although CT imaging usually has high resolution and signal to noise ratio, fully automatic segmentation of the heart is still a challenging problem due to similar intensity of neighboring organs and patient-specific anatomic variation [22],

and for this reason validation is a critical step in adoption of a new technique. Our atlas-based framework can be implemented to provide automatic delineations on each respiratory phase image, and these can be used to generate automatic planned organ at risks volumes for the heart.

In this work we implement our automatic whole heart segmentation algorithm on 4D-CT scans, and validate it by comparing manual and automated delineations. Moreover, we compare the union of its contours on all the respiratory bins with the contour of the heart on the MIP. Findings suggest that the proposed method is consistent with clinical contouring and can help facilitate automatic treatment planning for lung radiotherapy.

## Material and methods

This study received the approval from the Sichuan Cancer Hospital & Institute (Chengdu, China) ethics committee: approval number SCCHEC-02-2018-003. Patients gave verbal consent to the research. This retrospective study included the analysis of 4D-CT imaging datasets from 19 lung cancer patients who underwent radiotherapy in our Institution between December 2017 and January 2018; their characteristics are listed in Table 1.

Patients were immobilized in the supine position and fixed to the couch with a personalized thorax thermoplastic mask, arms crossed over the head. ANZAI belt systems (Anzai Medical Co. Ltd., Tokyo, Japan) were wrapped around the abdomen during the image acquisition. A 3D free breathing CT scan and subsequent 4D-CT scan, both with a slice thickness of 1.5 mm, were acquired with a Philips Big Bore CT scanner (Philips, Eindhoven, The Netherlands) with respiratory gating using the AZ-733VI system (Anzai Medical Co. Ltd, Tokyo, Japan) to measure the respiratory motion [23]. The 3D-CT and 4D-CT image sets (binned into ten breathing phases: 0% - 90%) were exported to MIM Maestro (MIM Software, Cleveland, OH, USA) for delineation by one of four experienced radiation oncologists, who contoured the heart in axial slices. Following the standard clinical routine at our hospital, target volumes were defined using all ten breathings phases, while organs at risk (OARs) were defined using only the 3D-CT scan. The 3D-CT image sets and contours were then imported into the Pinnacle 3TM Version 9.10 (Philips Medical Systems, Eindhoven, Netherlands) treatment planning system. For this research, the radiation oncologists were also asked to contour the heart volume (Heart$_{man}$) on each of the ten breathing phases. Additionally, the 4D-CT images were transferred to Monaco V5.5 (Elekta AB, Stockholm, Sweden) treatment planning system to create the MIP CT image dataset on which manual contouring of the planning organ at risk volume (PRV$_{MIP}$) for the heart was performed.

**Table 1. Patient characteristics in this trial cohort.**

|  | NUMBER (%) |
|---|---|
| *GENDER* |  |
| Male | 13 (65) |
| Female | 7 (35) |
| *AGE (YEARS)* |  |
| $\geq 60$ | 12 (60) |
| $< 60$ | 8 (40) |
| *TUMOR TYPE* |  |
| NSCLC (Squamous) | 6(30) |
| NSCLC (Adenoarcinoma) | 12 (60) |
| SCLC | 2 (10) |

The open-source, atlas-based automatic segmentation framework, previously developed by our team [20, 21], was used to automatically delineate the heart volume (Heart$_{auto}$) on the same image datasets using a leave-one-out approach. First, the heart was automatically segmented on the 0% phase image using multi-atlas segmentation, with the remaining patient images (at 0% phase) as the atlas set. Image registration was accomplished with a two-step registration process comprising initial rigid alignment and subsequent demons-based deformable image registration. This process also includes a procedure to crop the imaging volume to a volume of interest surrounding the heart, based on an automatic, intensity-based segmentation of the lungs. An atlas selection algorithm was then performed to automatically reject discordant atlases based on large surface deviations from a consensus delineation. This step does not rely on any manual contouring. From this reduced atlas set, local-weighted label fusion was used to generate Heart$_{auto}$ on the 0% phase image. Next, this automatic delineation was propagated to each of the remaining nine respiratory phase images using a separate intra-patient deformable image registration step. Lastly, the automatic PRV (PRV$_{auto}$) for the heart was defined as the union of the delineations on all breathing phase images. All automatic segmentations were validated by the radiation oncologists.

Quantitative metrics were used for each patient to compare Heart$_{man}$ and Heart$_{auto}$ for each breathing phase and PRV$_{auto}$ and PRV$_{MIP}$. We reported the Dice similarity coefficient (DSC), the mean absolute surface-to-surface distance (MASD), and the Hausdorff distance (HD; the maximum surface-to-surface distance). The DSC provides a measure of overlap between the automatic and manual delineations, with 0 indicating no overlap and 1 indicating perfect overlap. The MASD and HD are indicative of deviations between the delineations on the surface; with the HD being more sensitive to local surface deviations.

To validate the performance of our automatic segmentation more objectively, we also applied our proposed method independently to each phase of the binned 4D-CT images, since manual contouring was also used to delineate the heart independently in this way. We again use the DSC, MASD, and HD to evaluate the similarity between automatic segmentations and manual contours.

## Results

Overall, automatic segmentation of the whole heart volume is consistent with clinical practices, with minor deviations from manual contours, as seen for a representative patient in Fig 1. In Table 2, the results of DSC, MASD, and HD for the comparison of the automatic whole heart segmentation to manual contouring are reported. Results are given as the mean ± standard deviation. The cohort-averaged MASD indicates typical surface deviations on the order of voxel size, with no considerable outliers (min. 1.07 mm, max. 3.72 mm).

Generation of the PRV$_{auto}$ was also consistent with PRV$_{MIP}$, with a slightly higher HD than the comparisons between delineations on individual phases. In Fig 2, an example of a comparison of PRV$_{auto}$ and PRV$_{MIP}$ demonstrates the trend for the automatic algorithm to produce smoother and more anatomically consistent PRVs.

Comparing our proposed method to automatic segmentation performed independently on each phase of the binned 4D-CT images, we found little difference in the delineations (Fig 3). Analysis of quantitative metrics, as illustrated in Fig 4, demonstrated our proposed method to perform as well as this independent automatic segmentation. Minor differences were observed for individual patients, however aggregated results show no statistically significant differences in accuracy.

In a comparison of automatic and manual delineations for the best, median, and worst performing patient image sets (based on a ranking using the MASD), we demonstrate the high level of consistency provided by our proposed method (Fig 5). Segmentation accuracy is stable both between patients and across the binned respiratory phases from 4D-CT imaging.

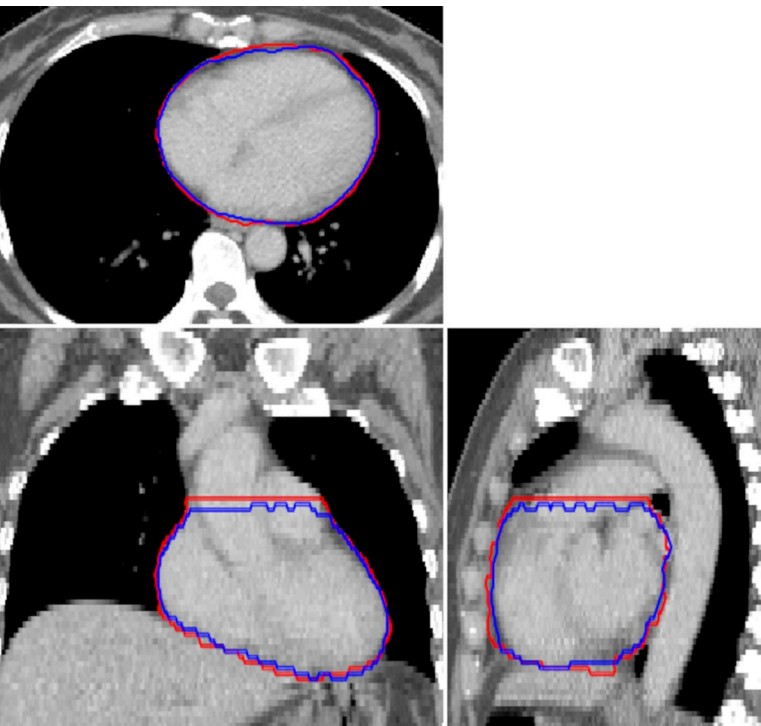

**Fig 1. Representative example of an automatic segmentation (blue) compared to manual contouring (red) of the whole heart.** The imaging volume shown corresponds to the automatically cropped region, defined using the bounding box of the lungs.

## Discussion

Manual contouring of 4D CT imaging is time-consuming and would thus greatly benefit from the use of automated techniques. This work outlines the application of an atlas-based segmentation framework to automatically delineate the heart in lung cancer radiotherapy planning

**Table 2. Quantitative comparison of the automatic whole heart segmentation to manual contouring.**

| Phase | DSC | MASD [mm] | HD [mm] |
|---|---|---|---|
| 0% | 0.934 ± 0.017 | 1.845 ± 0.514 | 12.896 ± 3.554 |
| 10% | 0.933 ± 0.017 | 1.893 ± 0.539 | 13.289 ± 3.602 |
| 20% | 0.935 ± 0.021 | 1.774 ± 0.625 | 12.540 ± 3.723 |
| 30% | 0.933 ± 0.022 | 1.869 ± 0.692 | 12.416 ± 3.651 |
| 40% | 0.934 ± 0.020 | 1.861 ± 0.667 | 12.619 ± 3.901 |
| 50% | 0.934 ± 0.021 | 1.859 ± 0.677 | 13.049 ± 4.034 |
| 60% | 0.931 ± 0.029 | 1.882 ± 0.823 | 12.655 ± 3.995 |
| 70% | 0.934 ± 0.020 | 1.867 ± 0.672 | 12.668 ± 3.837 |
| 80% | 0.932 ± 0.022 | 1.951 ± 0.781 | 13.192 ± 2.997 |
| 90% | 0.936 ± 0.019 | 1.834 ± 0.598 | 13.859 ± 6.882 |
| Average (all phases) | 0.934 ± 0.021 | 1.864 ± 0.666 | 12.924 ± 4.165 |
| PRV | 0.929 ± 0.014 | 2.074 ± 0.414 | 18.449 ± 8.496 |

DSC: Dice similarity coefficient, MASD: mean absolute surface-to-surface distance, HD: Hausdorff distance (maximum surface-to-surface distance). Results are given as the mean ± standard deviation (over 19 patients in the dataset).

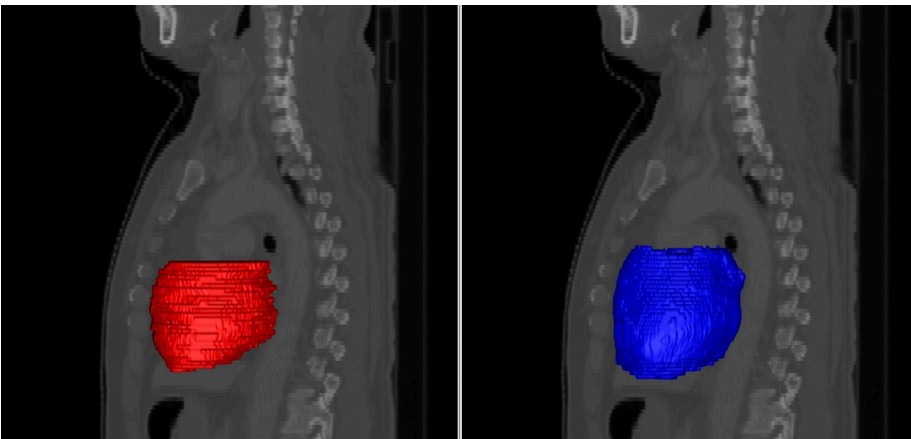

**Fig 2. Planning organ at risk volumes (PRVs) for the heart, generated by manually contouring the maximum intensity projection image manually (left) and the union of automatic segmentations on each breathing phase (right), shown here as surface meshes in sagittal projection.** Our automatic approach typically generates smoother and more anatomically consistent delineations.

using non-contrast 4D-CT imaging. We have shown that with a small dataset it is possible to reach clinically acceptable contouring accuracy, which makes this an attractive framework in situations where it may not be possible to gather sufficient data for other automatic segmentation approaches.

Automatic segmentation of the 0% phase image took approximately 20 minutes per patient, with deformable image registration making up the majority of this. Propagation of automatic segmentations to the remaining nine respiratory phases took approximately 5 minutes per patient, and the generation of $PRV_{auto}$ required less than a minute. This software was run on an 8-core CPU, and a decrease in execution time would be expected with more powerful hardware. Nevertheless, this is already an improvement considering the time taken to manually contour the heart volumes is typically 3–5 minutes for each respiratory phase, with a resulting total of approximately 30–50 minutes for the 4D-CT. The performance of this segmentation algorithm is similar to that of other research groups using an atlas-based approach for cardiac segmentation [24, 25], and to observed inter-observer contouring variability presented in the literature [20, 24, 26]. Importantly, there were no observed outliers in segmentation performance, indicating consistency with clinical practices and robust operation. We found that our proposed method was as accurate as applying our automatic segmentation algorithm independently to each phase image, with the benefit of a greatly reduced computational burden.

The primary aim of this work it to create a method that is suitable for clinical implementation within the context of automatic image segmentation for 4D lung cancer radiotherapy planning, with the goal of reducing the clinical workload associated with this technique. There are several practical challenges associated with this implementation. First, the intercommunication between the treatment planning system (TPS) and our software. Due to limitations on scripting functionality within the TPS, it is not possible to implement our software natively, which is also not ideal as this would make further generalizability difficult. Instead, we make use of a server-based solution which uses HTTPS and DICOM communication protocols to facilitate data transfer from the hospital picture archiving and communication system (PACS) to a separate server running our automatic segmentation software. After processing, the segmentations are returned to the PACS and can be loaded into the TPS for use in the clinical workflow. Ongoing pilot studies have demonstrated the feasibility of this approach, which

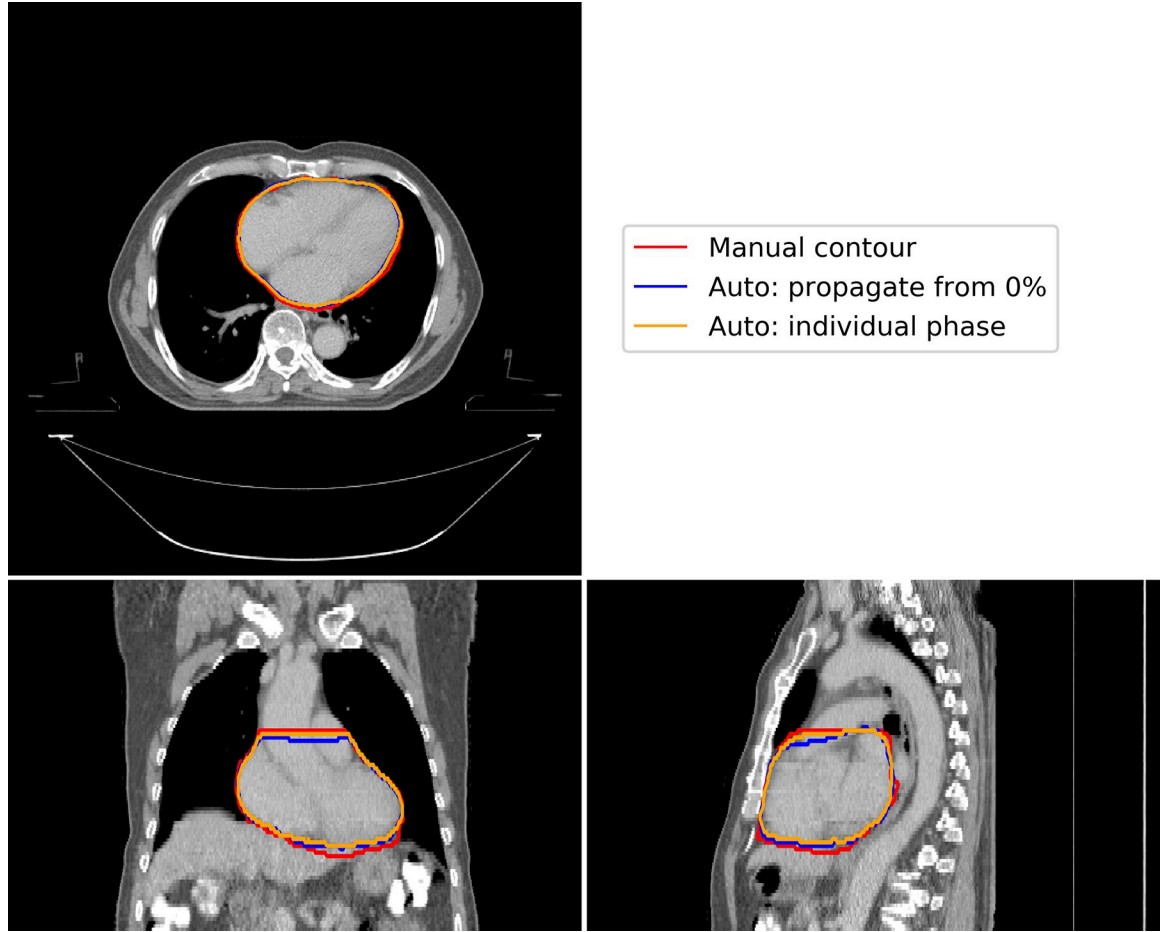

**Fig 3. A representative example of the differences in automatic segmentation using our proposed method (blue contour), propagation of delineations from the 0% phase using deformable image registration, and independent automatic segmentation for each individual phase (orange contour), compared to manual contouring (blue contour).** This figure illustrates results for the 40% phase image, where maximum deviation would be expected. There are minor variations at the base and apex of the heart.

provides a flexible and extensible platform. Another challenge for clinical implementation is reproducibility, where imaging of variable quality is a consideration. The image registration on which our atlas-based approach relies was designed to be robust to variability in image quality, and our segmentation algorithm has successfully been applied to large and heterogeneous radiotherapy imaging datasets [27]. While it is possible that imaging artefacts, for example from metallic implants or out-of-field effects, would affect image registration and hence increase the chance of a segmentation error, such artefacts would likely impact any automatic segmentation technique and would require manual intervention. We plan on using this software in a step between patient simulation and planning, and although we did not detect any segmentation errors in this dataset, output from our software will undergo clinician review prior to treatment planning. Ongoing research to evaluate our software in the clinical setting will provide data on the performance when considering potential variation in image quality.

As part of integration into the clinical workflow, we will assess the dosimetric impact of using our automatic approach. Initial investigation for a single patient, as shown in Fig 6, highlights minor differences between the clinical heart contour and the automatically defined heart volume. This patient was selected based on the shortest heart-tumour distance in this patient

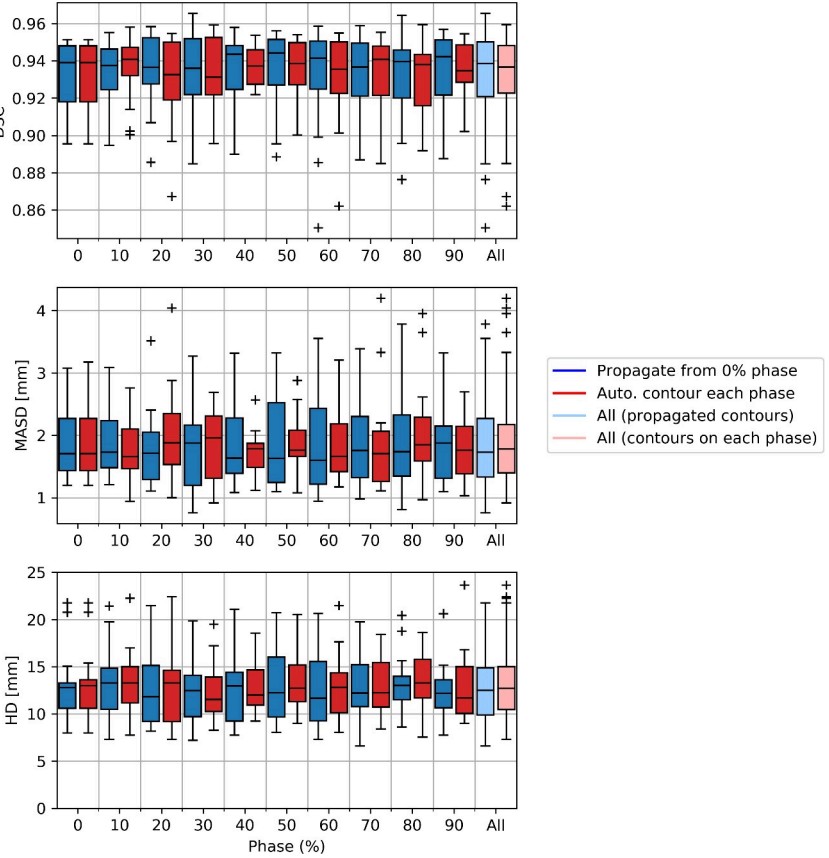

**Fig 4. A comparison on automatic segmentation using our proposed method, propagation of delineations from the 0% phase using deformable image registration, compared to independent automatic segmentation for each individual phase.** We use the Dice similarity coefficient (DSC), mean absolute surface-to-surface distance (MASD) and Hausdorff distance (HD) to compare the automatic segmentations derived using either method to the manual contouring. Results are presented as box plots aggregated over the patient cohort. Overall, both methods provide similar performance, with no statistically significant difference in delineation accuracy.

cohort, and thus represents the clinical situation for which the largest differences in dosimetry are likely to be observed. We find that relative to the clinical heart contour, the dose to the automatically defined heart volume is higher. As our automatic method defines the $PRV_{auto}$ as the union of delineations from each individual phase, it is feasible that the resulting volume captures more of the potential range of the heart due to respiratory motion. This highlights the importance of incorporating 4D treatment planning in lung cancer, and other thoracic tumours in general. In the future, we plan to extend this analysis to a larger patient cohort.

In current clinical radiotherapy practice for lung cancer, the only cardiac structure considered is the whole heart volume, with dose constraints used to limit the radiation exposure and subsequent cardiotoxicity risks. Given the focus of this study of establishing a system to facilitate automatic segmentation in the clinical workflow, we therefore validated our framework to delineate the whole heart volume only. It is worth noting that a growing body of research has demonstrated that whole heart dose parameters may not adequately predict cardiotoxicity risks [28–30]. In the future, it is possible that other cardiac structures will be routinely included in the radiotherapy planning process. Since our framework has also been shown to delineate many substructures with accuracy comparable to inter-observer contouring variability [20], this would make it feasible to extend our implementation to include these structures.

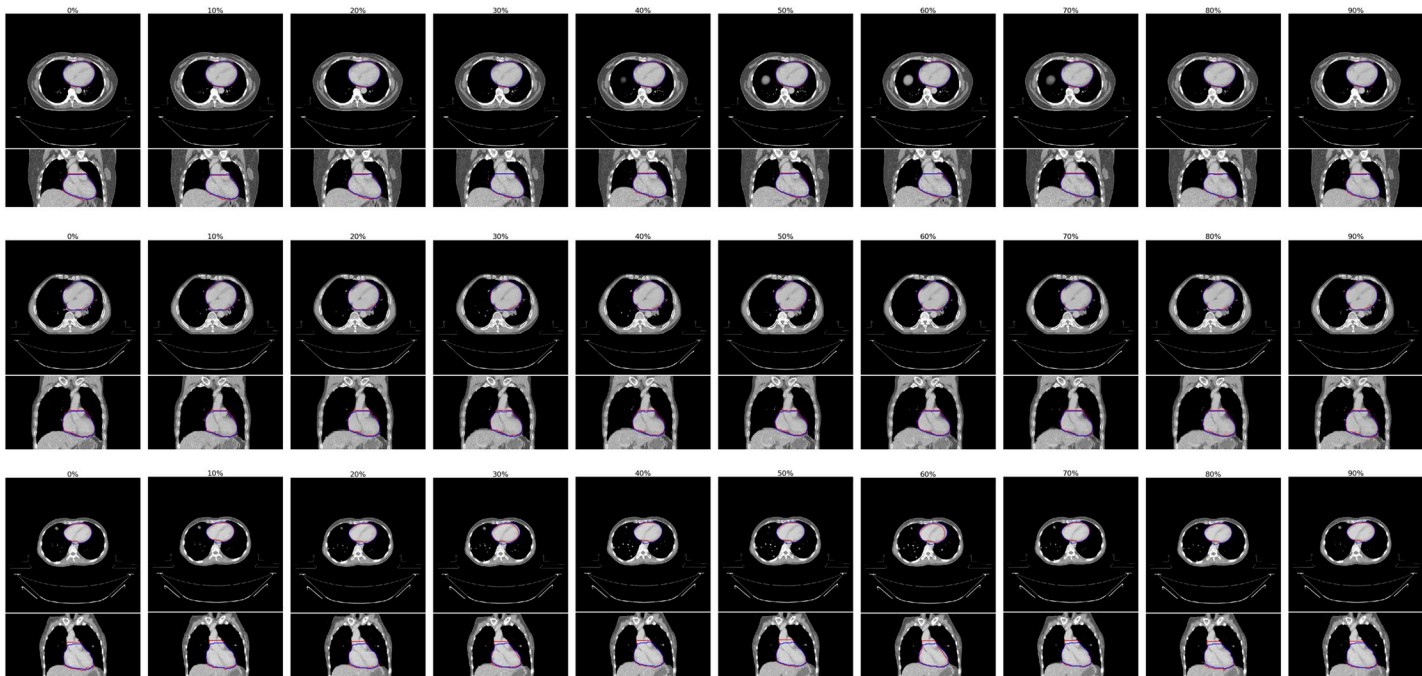

**Fig 5. Examples of results from our proposed automatic heart segmentation method (blue), compared to manual contouring (red), shown here for the best (top), median (middle) and worst (bottom) performing patient image sets, as ranked using the mean absolute surface-to-surface distance (MASD).** We display each binned respiratory phase image, 0% - 90% in 10% increments going left to right. Overall, our method provides consistently accurate delineations. Some variations from manual contouring can be seen, for example in the 60% in the bottom row, where deviations in manual contouring can be seen on the borders of the heart.

Additionally, retrospective analysis of the link between doses to cardiac substructures and patient outcomes, made possible with the use of our tool, could provide further evidence to better understand the link between cardiac radiation exposure and risks of long-term cardiotoxicities.

While deep learning approaches are quickly outpacing atlas-based techniques in terms of execution speed and accuracy, they often require large training datasets and are developed and validated on homogeneous imaging data. Many recent advances in cardiac segmentation using deep learning are demonstrating the potential for clinically acceptable performance delineation of cardiac magnetic resonance imaging (MRI) and cardiac CT imaging [31–35]. These imaging data, when compared to non-contrast radiotherapy planning CT imaging, are high-resolution, high-quality, and much more homogeneous. There is a growing body of work demonstrating the feasibility of deep learning methods for segmentation of radiotherapy imaging. Choi et al. (2020) [36] compared clinically-available atlas-based segmentation methods to an independently developed deep learning method for radiotherapy volumes contoured in breast cancer radiotherapy, including the heart. While their results demonstrate the superiority of their deep learning method, the use on contrast-enhanced CT imaging makes a comparison to this work difficult. The work of Morris et al. (2020) [37] presents promising results, with deep learning providing major gains in efficiency and accuracy for heart segmentation. Although results for the whole heart volume are not provided, their proposed method was able to delineate the cardiac chambers with a DSC of approximately 0.87 and a MASD below 2mm. While inference is performed on a standard radiotherapy planning CT, the network was trained using combined CT/MRI, data that is not typically available in retrospective clinical data. Interestingly, training this segmentation model only required 25 patient images, although the

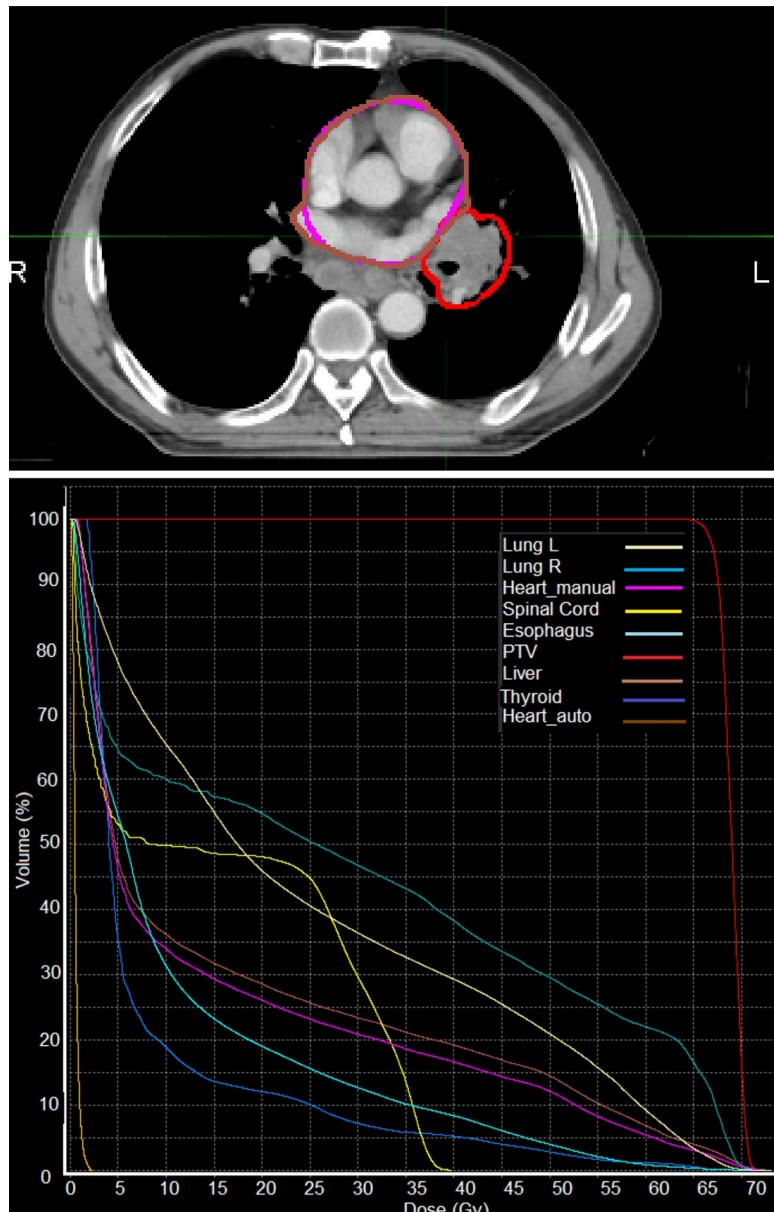

**Fig 6. A demonstration of an evaluation of the dosimetric impact of using the automatic heart segmentation (Heart_auto, brown) relative to the clinical heart contour (Heart_manual, pink).** We observe a higher overall dose to the automatically defined volume, which was generated as the union of delineations on each individual binned respiratory phase from the 4D-CT image. Top: contours for the heart and tumour (red) shown in an axial slice on the maximum intensity CT image, bottom: a dose-volume histogram for the planning target volume (PTV) and several organs at risk.

dataset represents consistent and homogenous imaging data which might not reflect clinical variability. Recent research has demonstrated the potential for accurate heart segmentation using standard, non-contrast radiotherapy planning CT imaging [38]. This work, aiming to reduce the clinical workload in adaptive planning approaches, suggests deep learning is indeed a feasible solution, with sub-second inference times and only minor manual adjustments required for clinical use. Their model was able to delineate the whole heart volume with a DSC

of 0.951, MASD of 2.31 mm, and HD of 8.5 mm, making their results comparable to this work. It is worth noting that their network was trained on using the Jaccard loss, which has a simple mathematical relation to the DSC, suggesting potential bias in the DSC scores.

Deep learning is poised to transform automatic segmentation, however, for this work, there are many reasons why an atlas-based was deemed more suitable. Fundamentally, our software is far more generalizable. It has been used to delineate the heart in radiotherapy planning imaging with variable image resolution, with imaging acquired using variable patient setup (e.g. both or single arm raised), and with known artefacts (e.g. from motion), without any manual processing or image resampling required. This makes our approach suitable for adoption in a clinical setting, where heterogeneous imaging data is a challenge for automating segmentation when using only a small dataset. Additionally, our method can be easily adapted to other clinical sites without any tuning of our algorithms, making it an attractive solution that can be implemented using clinically available data. As deep learning methods mature and become more accessible, we would expect that our accurate and algorithm will be useful alongside such methods for independent validation and quality assurance.

Our study had some limitations. Without a baseline measure of inter-observer variability, we relied on the comparison between the single manual contour and automatic segmentation to evaluate delineation consistency. Additionally, results should be interpreted in the context of the single-institute cohort used in this study. The accuracy of our atlas-based approach relies on defining anatomical correspondences between different patient images. Although an automatic atlas selection algorithm was used, large differences in patient anatomy relative to the atlas set may still result in lower registration accuracy. Furthermore, the 4D-CT imaging binned into respiratory phases, with the lack of cardiac gating, which may not be associated with or fully capture cardiac motion. This effect is likely to be minor, however, for tumours very close to the heart volume this might have an impact and warrants further investigation.

This work demonstrated the geometric consistency of automatic segmentation relative to clinical practices, with future work planned to ensure the doses calculated from these segmentations are accurate. Since dosimetric evaluation is specific to the particular treatment, such an evaluation generally requires a large study cohort to reach a definitive conclusion. The use of automatic segmentation may also facilitate the use of additional structures in treatment planning. In the future, we hope to extend our framework to delineate other structures at risk during lung radiotherapy.

## Conclusion

We have validated the accuracy of an automatic segmentation method for the delineation of the heart in lung cancer radiotherapy planning. This method was applied to 4D non-contrast CT scans, and demonstrates the potential to reduce the workload while maintaining consistency in clinical contouring.

## Author Contributions

**Conceptualization:** Lucia Orlandini, Xiongfei Liao, Jun Yin, Jinyi Lang.

**Data curation:** Lucia Orlandini, Xiongfei Liao, Jun Yin, Jinyi Lang.

**Formal analysis:** Robert Neil Finnegan, Jason Dowling.

**Investigation:** Robert Neil Finnegan, Lucia Orlandini, Jason Dowling, Davide Fontanarosa.

**Methodology:** Robert Neil Finnegan, Jason Dowling.

**Project administration:** Davide Fontanarosa.

**Resources:** Lucia Orlandini, Davide Fontanarosa.

**Software:** Robert Neil Finnegan, Jason Dowling.

**Supervision:** Lucia Orlandini.

**Visualization:** Robert Neil Finnegan.

**Writing – original draft:** Robert Neil Finnegan.

**Writing – review & editing:** Lucia Orlandini, Jason Dowling.

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
