## [Decision Letter · Decision Letter 0]

2 Sep 2020

PONE-D-20-20670

Feasibility of using a novel automatic cardiac segmentation algorithm in the clinical routine of lung cancer patients

PLOS ONE

Dear Dr. Finnegan,

Thank you for submitting your manuscript to PLOS ONE. After careful consideration, we feel that it has merit but does not fully meet PLOS ONE’s publication criteria as it currently stands. Therefore, we invite you to submit a revised version of the manuscript that addresses the points raised during the review process.  In particular, both reviewers suggested adding a more complete review of related literature, as well as a description of the clinical impact of the proposed methods.  Reviewer 1 also had significant concerns about the evaluation.  

We look forward to receiving your revised manuscript.

Kind regards,

Dzung Pham

Academic Editor

PLOS ONE

Journal Requirements:

Reviewers' comments:

Reviewer's Responses to Questions

**Comments to the Author**

1. Is the manuscript technically sound, and do the data support the conclusions?

Reviewer #1: Yes

Reviewer #2: Partly

2. Has the statistical analysis been performed appropriately and rigorously? 

Reviewer #1: Yes

Reviewer #2: Yes

3. Have the authors made all data underlying the findings in their manuscript fully available?

Reviewer #1: Yes

Reviewer #2: Yes

4. Is the manuscript presented in an intelligible fashion and written in standard English?

Reviewer #1: Yes

Reviewer #2: Yes

5. Review Comments to the Author

Reviewer #1: The authors proposed an atlas-based automatic heart segmentation method to delineate the heart in 4D-CT images acquired from lung cancer patients. Although there are merits of this work, there are some changes must be done before it can be accepted:

1. The major question of this study is that how the proposed method working in clinical practice? How about the reproducibility? The various image quality might be a problem for the proposed method.

2. The literature review of the work is highly insufficient, and the authors missed the most recent deep learning based segmentation methods. The authors should cite the following publications with proper discussions.

Zhuang, Xiahai, et al. "Evaluation of algorithms for multi-modality whole heart segmentation: an open-access grand challenge." Medical image analysis 58 (2019): 101537.

Yang, Guang, et al. "Simultaneous left atrium anatomy and scar segmentations via deep learning in multiview information with attention." Future Generation Computer Systems 107 (2020): 215-228.

Chengjia Wang et al. A Two-stage U-Net Model for 3D Multi-class Segmentation on Full-resolution Cardiac Data. The 9th Workshop on Statistical Atlases and Computational Modelling of MICCAI 2018, pp. 191--199.

Zenglin Shi et al. Bayesian VoxDRN: A Probabilistic Deep Voxelwise Dilated Residual Network for Whole Heart Segmentation from 3D MR Images. The 21st International Conference on Medical Image Computing and Computer Assisted Intervention (MICCAI 2018), pp. 569--577.

Yuanhan Mo et al. Deep Poincare Map: A Novel Method Coupling Dynamical System with Deep Learning for Left Ventricle Segmentation. The 21st International Conference on Medical Image Computing and Computer Assisted Intervention (MICCAI 2018), pp. 561--568.

3. Ideally, the authors should consider to compare with some deep learning based methods or at least have some discussions.

4. Why the automated results represent lots of ragged artefacts?

Reviewer #2: The proposed study investigates using an atlas-based cardiac segmentation method in the context of 4D-CT for delineation of the heart. The manuscript is clear, concise, and well written.

The paper does not propose a novel method, but rather a somewhat different application of a previously published method. I have a couple of major concerns:

1) The auto-contouring method was used only for contouring on the zero-phase and propagation to other phases was done via deformable image registration. This method of contour propagation is acceptable and widely performed in RT, however since the authors are comparing the proposed method to manually contoured structures (contoured on each phase), this leads to a biased comparison. Furthermore, the authors are essentially testing the feasibility of using 4DCT and DIR to propagate contours (which has been widely published on) rather than using the automated segmentation framework. The proposed method should either be applied independently to each breathing phase or the manual contours on the 0% phase should be propagated via DIR to the other phases. Since deformation vector fields must typically satisfy a smoothness constraint, this may also explain why contours generated via the atlas-based and DIR method are smoother than manual contours.

2) Many recent studies have posited that dose to the whole heart may not be adequate for correlating radiation dose with cardiac toxicity outcomes:

-Hoppe, Bradford S., et al. "The meaningless meaning of mean heart dose in mediastinal lymphoma in the modern radiation therapy era." Practical radiation oncology 10.3 (2020): e147-e154.

-McWilliam, Alan, et al. "Novel Methodology to Investigate the Effect of Radiation Dose to Heart Substructures on Overall Survival." International Journal of Radiation Oncology* Biology* Physics (2020).

-Stam, Barbara, et al. "Dose to heart substructures is associated with non-cancer death after SBRT in stage I–II NSCLC patients." Radiotherapy and Oncology 123.3 (2017): 370-375.

Given that previous publications by the authors have shown that the auto-segmentation method can be applied to cardiac substructures, in addition to the whole heart, I would suggest extending the analysis in the manuscript to include contouring of cardiac substructures.

3) The manuscript is very brief, and the immediate impact of the proposed method in the clinic is not clearly highlighted. Additionally, visual results from only one or two patients are shown. It would be helpful to include the segmentation results of all 10 phases on one or two patient slices.

Minor comments/concerns:

1) While some deep learning methods for automatic segmentation require large datasets, the method published by Morris et al:

Morris, Eric D., et al. "Cardiac substructure segmentation with deep learning for improved cardiac sparing." Medical physics 47.2 (2020): 576-586.

required only 25 patient datasets for training, so the authors' claim may be unfounded.

2) As the authors mention in the discussion - large datasets may need to be used to reach definitive conclusions on the dosimetric impact of the proposed method, some study on the dosimetric impact should be included. A comparison in heart DVH based on contours from the proposed method, and manual contours on the MIP and average CT would be helpful.

3) The cardiac contours are propagated via respiratory-correlated 4DCT, which may not be associated with or fully capture cardiac motion. Given this is standard practice in radiotherapy clinics, it is not a major concern, but the authors should at least mention this in the discussion section.

6. PLOS authors have the option to publish the peer review history of their article (what does this mean?). If published, this will include your full peer review and any attached files.

Reviewer #1: No

Reviewer #2: **Yes: **Joe Harms

---

## [Author Response · Author response to Decision Letter 0]

25 Nov 2020

Reviewer responses

We would like to extend our sincere thanks to the reviewers for providing constructive comments and thoughtful suggestions, which we feel have contributed to an improved manuscript. Overall, this has produced a more detailed and thorough description of both our work and the clinical and research context in which it is placed. Below we address each comment individually.

Reviewer 1

Comment 1. The major question of this study is that how the proposed method working in clinical practice? How about the reproducibility? The various image quality might be a problem for the proposed method.

Response 1. Thank you for this question. We agree that this is a primary concern. We have provided additional details regarding how we will implement our method in clinical practice, and discuss the challenge faced with variable image quality (Discussion - paragraph 3). 

Comment 2. The literature review of the work is highly insufficient, and the authors missed the most recent deep learning based segmentation methods. The authors should cite the following publications with proper discussions. (citations follow)

Response 2. We greatly appreciate the feedback and have taken this into consideration. We agree that many recent advanced in deep learning show promise for translation into the clinic, and have discussed the suggested published articles, along with several more, in detail in the revised manuscript (Discussion - paragraphs 6 and 7).

Comment 3. Ideally, the authors should consider to compare with some deep learning based methods or at least have some discussions.

Response 3. Thank you again for this suggestion. Following from comment 2, we have also provided a comparison to recent relevant research for cardiac delineation. 

Comment 4. Why the automated results represent lots of ragged artefacts?

Response 4. Thank you for this question. Using a multi-atlas approach, the automatic segmentation is essentially spatially averaged in 3D. Since tomographic imaging is reconstructed in axial planes, these segmentations can look jagged when viewed in 2D slices as a results of these factors. In three dimensions, automatic segmentations often look much smoother (as shown in Figure 2), which was a point raised by reviewer 2.

Reviewer 2

Comment 1. The auto-contouring method was used only for contouring on the zero-phase and propagation to other phases was done via deformable image registration. This method of contour propagation is acceptable and widely performed in RT, however since the authors are comparing the proposed method to manually contoured structures (contoured on each phase), this leads to a biased comparison. Furthermore, the authors are essentially testing the feasibility of using 4DCT and DIR to propagate contours (which has been widely published on) rather than using the automated segmentation framework. The proposed method should either be applied independently to each breathing phase or the manual contours on the 0% phase should be propagated via DIR to the other phases. Since deformation vector fields must typically satisfy a smoothness constraint, this may also explain why contours generated via the atlas-based and DIR method are smoother than manual contours.

Response 1. We greatly appreciate this in depth comment and the suggestion to further validate our methodology. In response, we have applied our automatic segmentation software to each phase independently. Results are given in Figures 3 and 4, with additional description in the revised manuscript (methods - final paragraph, results - final 2 paragraphs, discussion - end of paragraph 2).

Comment 2. Many recent studies have posited that dose to the whole heart may not be adequate for correlating radiation dose with cardiac toxicity outcomes [reference list]. Given that previous publications by the authors have shown that the auto-segmentation method can be applied to cardiac substructures, in addition to the whole heart, I would suggest extending the analysis in the manuscript to include contouring of cardiac substructures.

Response 2. Thank you for this suggestion. Firstly, we agree that much more research is required to better understand the link between radiation dose to cardiac substructures and cardiotoxicities. This is an area of research we are actively involved with (see our recent work published in Radiother. And Oncol.: https://doi.org/10.1016/j.radonc.2020.09.004).

The aim of this work was to demonstrate the feasibility of clinical implementation, and as current clinical practice does not consider cardiac substructures, we decided this extra information was not relevant for this study. With better cardiotoxicity models utilising cardiac substructure doses, it is likely this will change in the future, and we hope our tool will be able to be used to its full potential. Outside of clinical radiotherapy, this software might be applied for retrospective analysis of cardiac substructure doses more widely, as it has been already. We have discussed these points in the revised manuscript (Discussion - paragraph 5)

Comment 3. The manuscript is very brief, and the immediate impact of the proposed method in the clinic is not clearly highlighted. Additionally, visual results from only one or two patients are shown. It would be helpful to include the segmentation results of all 10 phases on one or two patient slices.

Response 3. Thank you for this suggestion. We believe that the revised manuscript presents a more complete analysis, and a more thorough investigation. Following your suggestion, we have included visual results for the best, median, and worst patient segmentation results (Figure 5).

Minor comments/concerns.

Comment 1. While some deep learning methods for automatic segmentation require large datasets, the method published by Morris et al required only 25 patient datasets for training, so the authors' claim may be unfounded.

Response 1. We thank you for bringing this to our attention. In the revised manuscript, we have added a description of deep learning methods and a comparison between relevant studies and our work (Discussion - paragraphs 6 and 7). It is correct that the work by Morris et al required only 25 cases to train a network, however a direct comparison to our framework is difficult as the data used by their group comprised paired CT/MRI (which is typically not available). Despite this, we are excited by the feasibility of using smaller datasets to train deep learning models, and look forward to future work in this area, as typically most studies present training datasets comprising much numbers of patient images.

Comment 2. As the authors mention in the discussion - large datasets may need to be used to reach definitive conclusions on the dosimetric impact of the proposed method, some study on the dosimetric impact should be included. A comparison in heart DVH based on contours from the proposed method, and manual contours on the MIP and average CT would be helpful.

Response 2. We appreciate this suggestion. In the revised manuscript we have included a dosimetric investigation for a patient selected from our study dataset with the smallest heart-tumour separation. We present a qualitative analysis of the differences in dose (see Figure 6), which we discuss (Discussion - paragraph 4). 

Comment 3. The cardiac contours are propagated via respiratory-correlated 4DCT, which may not be associated with or fully capture cardiac motion. Given this is standard practice in radiotherapy clinics, it is not a major concern, but the authors should at least mention this in the discussion section.

Response 3. Thank you for this astute observation. We have raised this point in our revised manuscript (Discussion - end of paragraph 8)

---

## [Decision Letter · Decision Letter 1]

29 Dec 2020

Feasibility of using a novel automatic cardiac segmentation algorithm in the clinical routine of lung cancer patients

PONE-D-20-20670R1

Dear Dr. Finnegan,

We’re pleased to inform you that your manuscript has been judged scientifically suitable for publication and will be formally accepted for publication once it meets all outstanding technical requirements.

Kind regards,

Dzung Pham

Academic Editor

PLOS ONE

Additional Editor Comments (optional):

Reviewers' comments:

Reviewer's Responses to Questions

**Comments to the Author**

1. If the authors have adequately addressed your comments raised in a previous round of review and you feel that this manuscript is now acceptable for publication, you may indicate that here to bypass the “Comments to the Author” section, enter your conflict of interest statement in the “Confidential to Editor” section, and submit your "Accept" recommendation.

Reviewer #2: All comments have been addressed

2. Is the manuscript technically sound, and do the data support the conclusions?

Reviewer #2: Yes

3. Has the statistical analysis been performed appropriately and rigorously? 

Reviewer #2: Yes

4. Have the authors made all data underlying the findings in their manuscript fully available?

Reviewer #2: Yes

5. Is the manuscript presented in an intelligible fashion and written in standard English?

Reviewer #2: Yes

6. Review Comments to the Author

Reviewer #2: I am appreciative that the authors went to great lengths to fully address my comments. The manuscript now includes a full and thorough analysis and discussion and is more than acceptable for publication. My only minor comment is that the axial slices in figure 5 could be further zoomed in on the heart, but it is adequate in its current form.

7. PLOS authors have the option to publish the peer review history of their article (what does this mean?). If published, this will include your full peer review and any attached files.

Reviewer #2: **Yes: **Joseph Harms

---

## [Editor Report · Acceptance letter]

6 Jan 2021

PONE-D-20-20670R1 

Feasibility of using a novel automatic cardiac segmentation algorithm in the clinical routine of lung cancer patients 

Dear Dr. Finnegan:

I'm pleased to inform you that your manuscript has been deemed suitable for publication in PLOS ONE. Congratulations! Your manuscript is now with our production department. 

Kind regards, 

on behalf of

Dr Dzung Pham 

Academic Editor

PLOS ONE